# MAIL: Mixture of LoRA Experts for Adaptive Incomplete Multimodal Learning

## Abstract

In real-world multimodal applications, modality missing frequently arises due to device failures, network instability, or privacy concerns. Latest studies introduce prompt-based fine-tuning process to adapt models in such incomplete multimodel learning scenarios. However, most of these methods struggle from two aspects: (1) static prompts are modality missing-aware but instance-invariant, thereby constraining the model performance. (2) the complexity of prompts are coupled with the number of modalities, hindering their scalability. Different from existing prompt-based methods, we propose **M**ixture of LoRA Experts for **A**daptive **I**ncomplete Multimodal **L**earning, named **MAIL**. Specifically, We design the LoRa-based Mixture of Experts and insert them into the pre-trained model to achieve adaptive incomplete multimodal learning. By training on datasets containing randomly missing modalities, MAIL can adaptively select a fixed combination of LoRA experts based on the current modality missingness and data unique characteristics. Accordingly, the parameter complexity depends only on a hyperparameter controlling the total number of experts, effectively decoupling it from the number of modalities. Extensive experimental comparisons on three real-world datasets demonstrate that MAIL can effectively handle incomplete modality problems compared to 11 baselines.

## 1 Introduction

The rapid development of pre-trained multimodal models in recent years Kim et al. (2021); Radford et al. (2021); Fan et al. (2023); Shu et al. (2025) has catalyzed remarkable progress in multimodal learning, driving advancements in tasks such as cross-modal retrieval Wang et al. (2024), image captioning Liu et al. (2025), and multimodal sentiment analysis Wang et al. (2025). While recent advances in multimodal learning have achieved remarkable success, most studies implicitly assume the complete availability of all modalities during both model pre-training and downstream inference. However, in real-world scenarios, factors like device failures, unstable networks, and privacy concerns can result in missing modalities in model inputs. These factors can severely undermine the model's performance and robustness, causing a sharp decline in its performance and robustness Ma et al. (2022); Hazarika et al. (2022).

Currently, research on alleviating modality missingness can be broadly categorized into three approaches: generation-based methods Ma et al. (2021); Yuan et al. (2021); Woo et al. (2023), joint learning methods Zuo et al. (2023); Wang et al. (2023); Yao et al. (2024), and prompt-based methods Lee et al. (2023); Jang et al. (2024); Pipoli et al. (2025); Lang et al. (2025). Generation-based methods aim to synthesize the missing modality using the available ones, creating pseudo-modal data that allow multimodal tasks to proceed as if all modalities were present. Joint learning methods focus on optimizing shared representations across modalities during training, enabling the model to make reliable predictions even when some modalities are missing, by leveraging these common representations. Recently, fueled by the growing capabilities of pre-trained models and the advancement of parameter-efficient fine-tuning (PEFT) techniques Wang et al. (2022b); Xu et al. (2025); Hu et al. (2022); Cheng et al. (2025), an increasing number of researchers have turned to prompt-tuning as a strategy to address the challenge of modality missingness. These prompt-based approaches exploit the powerful prior knowledge of pre-trained models to effectively handle missing modality scenarios by optimizing the inserted prompts (shown in Figure 1). Compared to the previous two approaches,

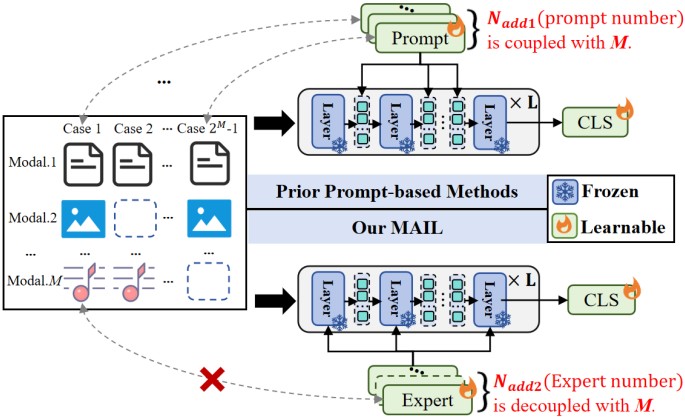

Figure 1: The comparison between existing prompt-based methods and our MAIL. Prompt-based method address modality-incomplete inputs by reconstructing missing modalities at the data level, inserting specific prompts into data embedding and intermediate hidden state for each modality-missing scenarios. MAIL, on the other hand, introduces trainable experts at the model level, adaptively selects experts combination according to the modality-missing scenarios as well as specific instance characteristics.

prompt-tuning can achieve competitive performance while significantly reducing computational and data requirements.

However, existing prompt-based methods typically face two problems: (1) Their prompts are statically shared across all input data under a fixed modality-missing scenario, ignoring the unique characteristics of individual data samples, which limits model performance. In other words, those prompts are pre-designed and changes only with the modality missing conditions, not with each input instance. (2) The number of prompt modules is coupled with the number of modalities. Since prompt-based methods operate directly at the data level, their complexity increases with the number of modalities. For example, with $M$ modalities, MAP Lee et al. (2023) requires predefined prompt parameters for all $2^M - 1$ possible missing combinations, while MSP Jang et al. (2024) reduces this to $M$, but still scales with modality count.

In view of the above, we introduce another PEFT method-Adapter and incorporate it with Mixture of Experts (MoE) to alleviate the existing problems. As shown in Figuer 1, instead of data-level prompt-based feature reconstruction, we insert Mixture of LoRA experts-a kind of low rank adapters, into the pre-trained model to achieve adaptively incomplete multimodel learning. Specifically, by fine-tuning on datasets with missing modalities, our method can adaptively select a fixed combination of LoRA experts based on the current modality missingness and data unique characteristics, thereby enhancing the model's inference performance and robustness. Accordingly, the parameter complexity depends only on a hyperparameter controlling the total number of experts, effectively decoupling it from the number of modalities. Besides, we also investigate different Moe designs, including linear-wise, attention-wise, and their combination, as well as different expert shape allocation strategies, to investigate their impact on modal missing scenarios, in order to further improve the performance of our proposed method. The main contributions of this paper could be summarized as follows:

- We introduce MAIL, which effectively mitigates the performance degradation caused by modality missing by adaptively selecting an optimal combination of experts, taking into account both the missing modalities and the characteristics of each sample.

- We investigate different MoE designs, including linear-wise, attention-wise, and their combination, as well as different routing strategies and expert allocation shapes, to investigate their impact under different modality-missing scenarios.

- We conduct extensive experiments on three real-world datasets to evaluate MAIL in comparison with 11 competitive baselines and the results confirm MAIL's effectiveness in addressing missing-modality issues.

## 2 RELATED WORK

**Multimodal Learning with Missing Modalities.** Existing approaches for incomplete multimodal learning can be categorized into three types: *(1) Generation-based methods* Ma et al. (2021); Yuan et al. (2021); Woo et al. (2023) aim to reconstruct the missing modalities. These methods often rely on generative models to infer the latent representation of the absent modality conditioned on the observed modalities. *(2) Joint learning methods* Zuo et al. (2023); Wang et al. (2023); Yao et al. (2024) focus on learning unified representations across modalities during training, with the goal of capturing intrinsic correlations of modalities. *(3) Prompt-based methods* Lee et al. (2023); Jang et al. (2024); Pipoli et al. (2025); Lang et al. (2025) have recently emerged as a more resource-friendly way to handle modality-missing problems. These approaches exploit the knowledge of pre-trained multimodel models and introduce few learnable prompts to encode modality-specific cues, enabling the model to adapt incomplete multimodel learning scenarios.

However, existing prompt-based methods often rely on static modality-missing-aware prompts that ignoring specific unique characteristics. Besides, their complexity of added prompts are coupled with the number of modalities. These two problems limit the performance and scalability of prompt-based methods.

**PEFT for pre-trained multimodel models** Common PEFT methods used in multimodal learning scenarios can be categorized into two types: prompts and adapters. CoOp Zhou et al. (2022b) and CoCoOp Zhou et al. (2022a) are the first foundational works that introduce prompts into multimodal models, which designs trainable prompts to encode the label of each image into a trainable context. MaPLe Khattak et al. (2023) extends trainable prompts that only exist on the text encoder to both text and image encoders. Prompt-ladder Cai et al. (2025) significantly reduces the memory usage of prompt-tuning through a lightweight ladder network to bypass large pre-trained models during back-propagation. There are also many studies that introduce adapters into multimodal models and achieve great success. CLIP-Adapter Gao et al. (2024b)adds adapters to the end of both the visual and textual branches of the CLIP model, enabling the model to learn new features for downstream tasks. Tip-Adapter Zhang et al. (2022) proposed a training-free approach that does not require training the adapter through backpropagation enables more efficient multimodal model transfer.

Recently, researchers Wang et al. (2022a); Zadouri et al. (2023); Yu et al. (2025) combine MoE with PEFT methods, offering new approaches for adapting pre-trained models to multi-task and other complex scenarios, and achieving notable performance improvements. Inspired by these works, we observe that modality-missing scenarios are naturally well-suited for fine-tuning via MoE based PEFT, offering a resource-efficient way to enhance model performance and robustness.

## 3 METHODOLOGY

### 3.1 PROBLEM STATEMENT

Assuming the multimodal dataset $D$ contains $M = 2$ modalities, denoted as $m_1$ and $m_2$, representing text and image modalities, respectively. Since all modalities missing are meaningless, the model needs to handle a total of $2^M - 1$ possible missing modality cases. Specifically, for dataset within two modalities (text and image), three modality-missing scenarios will arise: $D^c = \{(x_i^{m_1}, x_i^{m_2}, y_i)\}_1^{n_c}$, $D^{m_1} = \{(x_i^{m_1}, y_i)\}_1^{n_1}$, $D^{m_2} = \{(x_i^{m_2}, y_i)\}_1^{n_2}$, where $D^c$, $D^{m_1}$, and $D^{m_1}$ are the subset of different modality missing scenarios and $D = \{D^c, D^{m_1}, D^{m_1}\}$. $y_i$ denotes the i-th instance label. $x_i^{m_1}$ and $x_i^{m_1}$ are the instance of two modalities. $n_c$, $n_1$, and $n_2$ represent the total number of instances for different missing modality scenarios. In order to ensure input consistency, missing modalities will be supplemented with dummy data (eg, empty string for text, zeros for image) in subset $D^{m_1}$ and $D^{m_2}$, which results $\tilde{D}^{m_1} = \{(x_i^{m_1}, \tilde{x}_i^{m_2}, y_i)\}_1^{n_1}$ and $\tilde{D}^{m_2} = \{(\tilde{x}_i^{m_1}, x_i^{m_2}, y_i)\}_1^{n_2}$. The final multimodal dataset with missing modalities can be denote as: $\tilde{D} = \{D^c, \tilde{D}^{m_1}, \tilde{D}^{m_1}\}$, which is a mixture of subset with different modalities-missing scenarios.

Figure 2 provides an overview of our proposed MAIL. The primary objective of MAIL is to fine-tune experts integrated into the pretrained model under multimodal datasets with different modality missing scenarios. By jointly considering both the missing modality patterns and instance-specific features, MAIL dynamically selects an appropriate combination of experts, thereby enhancing the model's inference performance and robustness.

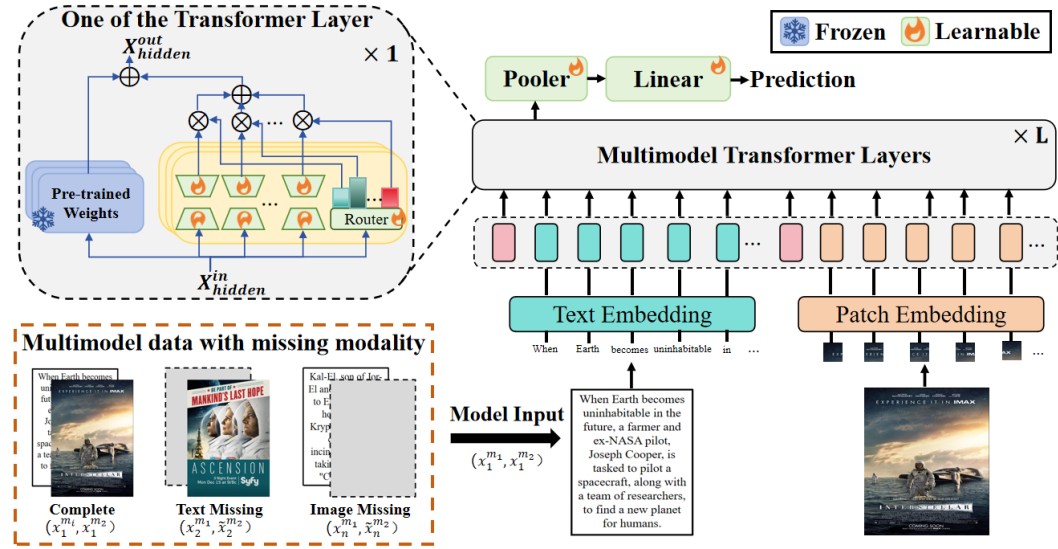

Figure 2: The overview of MAIL. By fine-tuning on datasets with missing modalities, MAIL can adaptively select a fixed combination of LoRA experts based on the current modality missingness and data unique characteristics, thereby enhancing the model's inference performance and robustness.

## 3.2 REVISITING ViLT

Given its proven effectiveness across a wide range of multimodal learning tasks, we adopt the pre-trained ViLT model Kim et al. (2021) as the backbone of our framework. Assume the current text and image inputs are denoted as $x_i^{m_1}$ and $x_i^{m_2}$, respectively. ViLT directly encode the text input $x_i^{m_1}$ into text embedding $t \in \mathbb{R}^{H_1 \times d}$ containing the $cls$ token and position embedding. Similarly, image input $x_i^{m_2}$ will first divided into patches, which are then flattened and passed through a linear projection to obtain the image embedding $v \in \mathbb{R}^{H_2 \times d}$. Then, text and image embeddings will be summed with their corresponding modality-type embedding vectors $e_t, e_v \in \mathbb{R}^d$, and then concatenated into a single sequence and fed into the L-depth transformer layers, the formulation are as follows:

$$h_i^0 = [t + e_t, v + e_v] \tag{1}$$

$$h_i^l = TransformerLayer^{(l)}(h_i^{l-1}), \quad l = 1, 2, ..., L \tag{2}$$

Finally, the resulting vector $h_i^l$ will be passed through a pooler, which applies a linear transformation $W_{pool}$ followed by a non-linear activation $tanh$. The output is then fed into a lightweight, task-specific classification head comprising a fully connected layer to get the model predictions.

$$\hat{y}_i = FC(tanh(h_i^l W_{pool})) \tag{3}$$

## 3.3 LOW-RANK ADAPTATION

Low-Rank adaptation (LoRA) is a adapter-like PEFT approach that is widely used in fine-tuning. It freezes the parameters of the pre-trained model and decomposes the parameter updates into low-rank matrices, thereby significantly reducing the number of trainable parameters. For the given i-th layer of pre-trained model with weight matrix $W_i \in \mathbb{R}^{d_{out} \times d_{in}}$, LoRA introduces two learnable low-rank matrices: $A \in \mathbb{R}^{r \times d_{in}}$ and $B \in \mathbb{R}^{d_{out} \times r}$, where $r$ is a small rank satisfying $r \ll \min(d_{in}, d_{out})$. This design ensures that the modified transformation $BAx$ maintains the same output dimensionality as the original $W_i x$. The computation of LoRA can be formulated as follows:

$$h = W_i x + \Delta W_i x = W_i x + \frac{\alpha}{r} BAx \tag{4}$$

where $r$ represents LoRA rank, $\alpha$ is the scaling factor that controls the change magnitude of $W_i$ .

In practice, $A$ is initialized with values drawn from a Gaussian distribution, while $B$ is initialized to zeros. This initialization strategy guarantees that the model's behavior at the start of fine-tuning remains identical to that of the original pre-trained model. Compared to full fine-tuning of all model weights, LoRA achieves a substantial reduction in trainable parameters, while still maintaining competitive performance levels.

### 3.4 MIXTURE OF LoRA EXPERTS

Traditional MoE architectures Shazeer et al. (2017); Fedus et al. (2022) enhance model capacity by substituting model's linear layers with specialized MoE layers. Each MoE layer comprises a set of $N$ independently parameterized linear layers, referred to as experts, denoted by $\{E_i\}_{i=1}^N$. A routing mechanism $R(\cdot)$ is employed to determine the contribution of each expert dynamically, based on the input. Given an input $x$, the MoE layer produces an output $y$ by aggregating the responses of all experts, each weighted by the router's output. The formulations are as follow:

$$y = \sum_{i=1}^{N} R(x)_i \cdot E_i(x) \tag{5}$$

$$R(x)_i = \text{Softmax}(W_r x) \tag{6}$$

Here, $E_i(x)$ represents the $i$-th linear expert's output, and $R(x)_i$ indicates the weight of associated router. $W_g$ is the trainable weight matrix for router. $R(\cdot)$ typically computes a probability distribution over experts conditioned on $x$, enabling soft selection or sparse routing depending on the design. After training with amount of data, traditional MoE architecture can leverage the router to enable each expert for realizing unique capabilities.

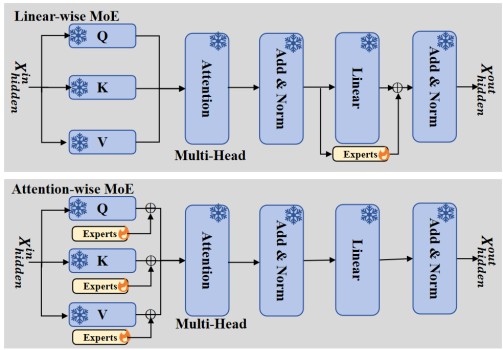

Unlike traditional MoE approaches that replace the model's linear layers with MoE layers containing multiple linear experts, we propose Mixture of LoRA Experts (MoRA), which uses multiple efficient LoRA experts to substitute the parameter changes during fine-tuning. This significantly

Figure 3: The illustration of two Mixture of LoRA Experts designs.

reduces the demand for computational resources and learning data. In this paper, We explore the two designs of the mixture of LoRA experts architecture: *Linear-wise MoRA* and *Attention-wise MoRA*.

#### 3.4.1 LINEAR-WISE MoRA.

A common designs that target at linear layers within the pre-trained model, as shown in top of the Figure 3. Suppose that the linear layer of the i-th transformer layer is defined as:

$$Linear(x)_i = W_i x + \Delta W_i x \tag{7}$$

By applying Linear-wise MoRA, the weight changes $\Delta W_i$ during fine-tuning will be substitute with low-rank format MoE layers, the new linear layer can be formulated as follows:

$$\hat{Linear}(x)_i = W_i x + \frac{\alpha}{r} \sum_{i=1}^{N} R(x)_i B_i A_i x \tag{8}$$

As previously mentioned, $r$ represents LoRA rank, $\alpha$ is the scalling factor, $R(x)_i$ indicates the router, $A_i$ and $B_i$ are the low-rank matrices.

#### 3.4.2 ATTENTION-WISE MoRA.

We also introduce another designs of MoRA that focus on MSA layers Vaswani et al. (2017), as shown in the bottom of Figure 3. We replace the weight changes of original query, key and value

with low-rank experts. Same as Linear-wise MoRA, the formulations can be defined as follow:

$$\hat{Q}(x)_i = W_i^q x + \frac{\alpha}{r} \sum_{i=1}^{N} R(x)_i^q B_i^q A_i^q x \tag{9}$$

$$\hat{K}(x)_i = W_i^k x + \frac{\alpha}{r} \sum_{i=1}^{N} R(x)_i^k B_i^k A_i^k x \tag{10}$$

$$\hat{V}(x)_i = W_i^v x + \frac{\alpha}{r} \sum_{i=1}^{N} R(x)_i^v B_i^v A_i^v x \tag{11}$$

Where $W_i^q, W_i^k, W_i^v \in \mathbb{R}^{d \times d}$ is the projection weights of attention layer. Then, the attention layers can be defined as:

$$\hat{Attention}(x) = Softmax \left( \frac{\hat{Q}(x)_i \hat{K}(x)_i^{\top}}{\sqrt{d}} \right) \hat{V}(x)_i \tag{12}$$

Since Attention-wise MoRA targets more objectives, it introduces more additional expert parameters compared to Linear-wise MoRA. In the experiments, the variant using linear-wise MoRA is named MAIL-l, and the one using attention-wise MoRA is named MAIL-a. Additionally, we designed a variant that incorporates both two MoRA, named MAIL-b.

### 3.4.3 OTHER MoRA SETTINGS.

In addition to the types of MoE designs, we also try different routing strategies and expert allocation shapes in the experiment, to investigate their impact under different modality-missing scenarios.

For routing strategies, we adopt Top-K strategy that select the most appropriate K experts for each layer in a discrete way. Besides, we also try a *SoftMerge* strategy Zadouri et al. (2023) that compute a weighted average of experts. The results show that simple Top-K is better than *SoftMerge*, as the latter strategy may introduce more noise in scenarios with different missing modalities.

For expert shape allocation, prior research Gao et al. (2024a) investigates the impact of different expert shape and find that a $\bigtriangledown$ allocation shape yields better performance when applied to pretrained models, which assigns more experts to higher layers (output side) of the model while allocating fewer experts to the lower layers (input side). Follow their settings, we apply different expert allocation shapes and construct MAIL-$\bigtriangledown$, MAIL-$\triangle$, and MAIL-$\square$. MAIL-$\bigtriangledown$ allocate more experts in the higher layers of pre-trained model. MAIL-$\triangle$ is opposite to MAIL-$\bigtriangledown$, which allocates fewer experts in higher layers. MAIL-$\square$ allocates the same number of experts across all layers of the pre-trained model. We find under modality-missing scenarios MAIL-$\square$ performs better than others. We will discuss in more details in the evaluations section.

## 4 EVALUATIONS

### 4.1 EXPERIMENTAL SETTINGS

**Datasets.** Following prior research Lee et al. (2023); Jang et al. (2024), we evaluate our MAIL across three multimodal downstream datasets: (1) *MM-IMDb* Arevalo et al. (2017), a benchmark dataset designed for movie genre classification that leverages both textual and visual information; (2) *UPMC Food101* Wang et al. (2015), which emphasizes image classification with complementary textual input; and (3) *Hateful Memes* Kiela et al. (2020), which targets hate speech detection within memes by integrating image and text modalities.

**Missing data setting.** In our experiment, the training and test set both contain modality-missing data. We define $\eta\%$ as the missing rate, which indicates the proportion of incomplete data pair within the multimodel dataset. Since the dataset contains two modalities (text and image), the modality missing types can be devided as: (1) text/image missing (denotes as **Text/Image** in all tables) indicates that there are $\eta\%$ image-only/text-only data pairs and $(1-\eta)\%$ complete data pairs within the dataset. (2) both missing (denotes as **Both** in all tables) indicates that there are $\frac{\eta}{2}\%$ text-only data, $\frac{\eta}{2}\%$ image-only data, and $(1-\eta)\%$ complete data. We can extend the modalities to $M$

in total. In this case, each missing type will have $(\frac{\eta}{M^2-2})\%$ incomplete data samples and $(1-\eta)\%$ complete data. In our experiments, the default value of missing rate $\eta$ is set to 70%.

**Evaluation Metrics.** We adopt different metrics for each dataset follow the common settings. For MM-IMDb Arevalo et al. (2017), F1-Macro is used to indicate the performance of multi-label classification task. For UPMC Food-101, accuracy is adopted that suitable for normal classification task. For Hateful Memes, we use Area Under the Receiver Operating Characteristic Curve (AUROC) to evaluate model's performance.

**Implementation Details.** Following prior worksLee et al. (2023); Lang et al. (2025), we utilize the pre-trained ViLTKim et al. (2021) as the backbone and the parameters of ViLT remain frozen during fine-tuning. Only inserted experts and downstream related pooler and linear layers need to optimize. AdamW optimizer is adopted with a learning rate of 2e-4. We set each LoRA expert with rank $r = 4$, $\alpha = 8$, and LoRA dropout is 0.05. The top-2 routing strategy is used as default. For default epxert allocation shape, we construct a square shape (□) MoE, which allocate 5 experts to all layers of ViLT. All experiments are conducted with an NVIDIA RTX 4090 GPU.

Table 1: Comparison of different methods across MM-IMDb, HateMemes, and UPMC Food-101 datasets. Text, Image, Both indicate different Missing Type with default 70% missing rate, as detailed in the Missing data setting section.

| Methods | MM-IMDb | | | Hateful Memes | | | UPMC Food-101 | | |
|---|---|---|---|---|---|---|---|---|---|
| Missing Type | Text | Image | Both | Text | Image | Both | Text | Image | Both |
| *Generation-based methods* | | | | | | | | | |
| SMIL | 38.32 | 27.57 | 35.12 | 50.32 | 58.50 | 54.63 | 51.83 | 49.86 | 46.77 |
| TFR-Net | 37.70 | 39.45 | 37.24 | 51.18 | 55.57 | 52.12 | 65.91 | 67.58 | 63.41 |
| AcMAE | 47.47 | 43.82 | 44.05 | 55.74 | 59.66 | 57.25 | 69.28 | 73.75 | 71.15 |
| *Joint learning methods* | | | | | | | | | |
| IF-MMIN | 39.63 | 31.95 | 31.98 | 57.62 | 53.44 | 55.19 | 66.76 | 64.36 | 68.53 |
| ShaSpec | 44.04 | 44.23 | 44.06 | 58.75 | 60.30 | 60.96 | 60.99 | 74.87 | 70.02 |
| DrFuse | 47.05 | 44.09 | 48.83 | 57.60 | 60.66 | 55.84 | 66.30 | 75.09 | 68.23 |
| CorrKD | 44.82 | 39.48 | 41.20 | 58.74 | 55.59 | 57.91 | 61.37 | 66.83 | 62.87 |
| *Prompt-based methods* | | | | | | | | | |
| MAPs | 46.12 | 44.86 | 45.48 | 58.62 | 60.16 | 58.89 | 67.02 | 75.62 | 72.52 |
| MSPs | 49.16 | 44.62 | 48.28 | 59.60 | 60.05 | 59.08 | 71.74 | 79.09 | 74.46 |
| SCP | 48.16 | 44.78 | 46.29 | 57.34 | 59.47 | 58.14 | 73.68 | 78.96 | 76.97 |
| RAGPT | 55.16 | **46.44** | 50.89 | **64.10** | 62.57 | **63.47** | 75.53 | **81.98** | 76.94 |
| MAIL-l | 54.97 | 46.07 | 51.27 | 62.26 | 61.71 | 61.95 | 74.26 | 80.36 | 76.58 |
| MAIL-a | 54.34 | 45.61 | 51.81 | 62.81 | 61.96 | 62.44 | 75.45 | 80.36 | 76.82 |
| MAIL-b | **55.42** | 46.39 | **52.06** | 63.47 | **62.73** | 63.11 | **75.77** | 81.26 | **77.28** |

## 4.2 MAIN RESULTS

We compare MAIL with 11 competitive baselines, which are classified into three categories: (1) *Generation-based methods*: SMIL Ma et al. (2021), TFRNet Yuan et al. (2021), and AcMAE Woo et al. (2023). (2) *Joint learning methods*: IF-MMIN Zuo et al. (2023), ShaSpec Wang et al. (2023), DrFuse Yao et al. (2024), and CorrKD Li et al. (2024). (3) *Prompt-based methods*: MAP Lee et al. (2023), MSP Jang et al. (2024), SCP Pipoli et al. (2025), and RAGPT Lang et al. (2025).

Table 1 shows an overall comparison between MAIL and 11 baseline methods across three datasets. We can observed that: (1) MAIL achieves the best or second-best performance in the majority of experimental settings compared to baselines, which demonstrating that select specific expert combinations informed by both modality-missing scenarios and instance-specific characteristics can significantly alleviate the impact of missing modalities. (2) Although MAIL-a that adopt attention-wise MoRA has more parameters, its overall performance across different experimental settings is comparable to that of the MAIL-l which adopt linear-wise MoRA. This is possibly because the linear operates on each token independently, making it naturally well-suited for the sparse expert selection mechanism of MoE, and thus it tends to perform better. (3) MAIL-b, which applies MoE at both the linear and attention-wise MoRA, undoubtedly achieves the best performance due to the

inclusion of more trainable parameters. (4) generation-based and joint learning methods exhibit worse performance, primarily attributable to the uncertainty induced by stochastic placeholders and the intrinsic challenges posed by cross-modal heterogeneity during the reconstruction process, both of which constitute substantial impediments to model efficacy. In addition, prompt-based methods also demonstrate limited performance, as most of their static prompting ignored important instance-specific features.

## 4.3 ABLATION STUDY

**Robustness to different missing rates.** We conduct further experiments to analyze the robustness of our proposed method against different missing-modality rates between training and testing phases. First, we use radar charts to clearly illustrate the performance of different methods under varying missing rates, shown in Figure. 4. ViLT experiences a significant decline in performance when any modality is absent during inference. In contrast, the figures indicate that MAIL maintains consistently strong performance across varying rates of missing data, highlighting the enhanced robustness of our approach.

Figure.5 presents the performance of different methods under more continuously varying missing rates. We can also observed that:(1) All methods naturally experience a gradual decline in performance as the missing rate increases. (2) Missing-both will cause greater fluctuations in model performance as the missing rate increases. (3) Benefiting from well-designed retrieval framework, RAGPT surpasses MAIL under several missing rate settings, but MAIL still demonstrates superior overall performance.

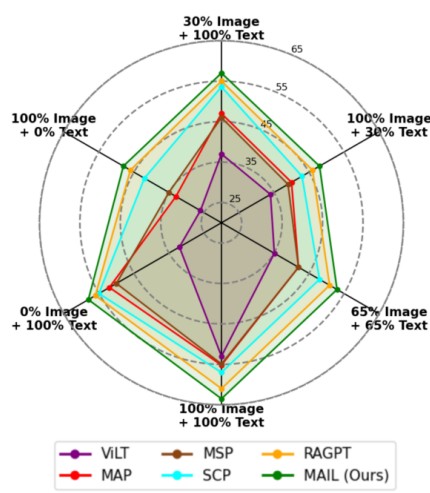

Figure 4: The radar plot shows F1-macro scores on the MM-IMDb dataset, with each axis representing the % availability of image and text modalities.

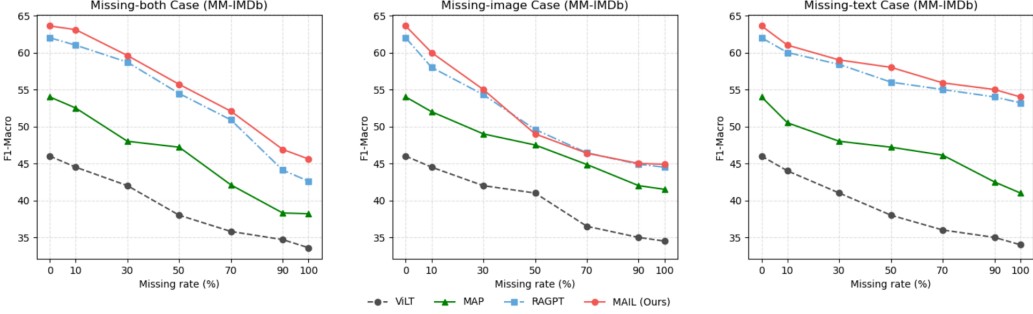

Figure 5: The comparison results of MAIL with several prompt-based methods on the MM-IMDb dataset with different missing rates (both training and test dataset with the same missing rate).

Finally, we further compare MAIL variants under different missing scenes follow prior work's setting Lee et al. (2023) to analyze the robustness of the model when the training set and test set belong to different modality-missing scenarios, results are shown in Figure.6. In Figure 5(a), all variants of MAIL are trained on data with the Both missing type and tested on the Text missing type. We can find that although MAIL-a has more updatable parameters, the gap between MAIL-a and MAIL-l gradually narrows as the missing rate increases, indicating that linear-wise MoRA may be more suitable for modal missing scenarios. In Figure 5(b), training is conducted on Both missing type data with 10%, 70%, and 90% missing rates, and the models are transferred to testing on data with various other missing rates. We observe that at lower modality missing rates, training with more modality-complete data yields improved model robustness and performance. However,

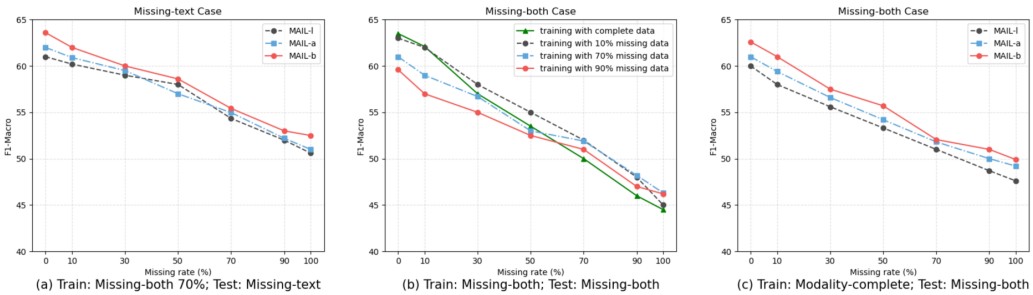

Figure 6: Ablation study on robustness to the testing missing rate in different scenarios on MM-IMDb.

as the missing rate increases, the influence of data completeness on model performance progressively diminishes. In Figure 5(c), training is performed entirely on modality-complete data, and the models are transferred to testing on data with different missing rates of Both missing type. Since the model has not encountered modality-missing scenarios during training, its performance degrades more significantly as the modality missing rate increases.

**The impact of different expert allocation shapes and routing strategies.** To further investigate the impact of different MoE configurations on modality-missing scenarios, we conducted ablation studies on both the routing strategies and expert allocation shapes.

Table 2 presents the results of different routing strategies. We find that the default top-2 strategy achieves the best performance. Although previous studies have shown that soft merge can perform well by integrating more experts, in scenarios with modality missing, experts may specialize in different modalities, and merging their outputs could introduce additional noise, ultimately leading to degraded performance.

Table 3 presents the results of different expert allocation shapes. Previous studies have found that an $\triangledown$ expert shape tends to yield better performance. However, our results show that a $\square$ expert shape performs better in scenarios with modality missing. We think that this is because missing modalities require the model to involve more experts on the input side during training to compensate for the missing information of original data.

Table 2: Analysis of routing strategies.

| Routing strategy | Text | Image | Both |
|---|---|---|---|
| Top-2 (Default) | **55.42** | 46.39 | **52.06** |
| Top-1 | 52.84 | 44.47 | 50.14 |
| *SoftMerge* | 54.63 | **46.52** | 51.38 |

Table 3: Analysis of expert allocation shapes.

| Expert allocation Shape | Text | Image | Both |
|---|---|---|---|
| MAIL-$\square$ (Default) | **55.42** | **46.39** | 52.06 |
| MAIL-$\triangle$ | 55.07 | 46.29 | 51.91 |
| MAIL-$\triangledown$ | 55.34 | 46.03 | **52.22** |

## 5 CONCLUSION

In this work, we propose MAIL, a Mixture of LoRA Experts for Adaptive Incomplete Multimodal Learning. Unlike existing prompt-based approaches that suffer from static prompts and scalability issues, MAIL dynamically selects expert combinations based on both modality-missing scenarios and instance-specific features, achieving greater adaptability and robustness. Through extensive experiments on three real-world datasets, MAIL outperforms 11 strong baselines in many cases, demonstrating its effectiveness and scalability in handling incomplete multimodal scenarios.

ACKNOWLEDGMENTS

This research project is supported by the National Natural Science Foundation of China (No.x)

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
