# Supplemental material

## 1.More detailed experimental setup

**Datasets.** We follow the experimental protocol defined by Lee et al. and report experiments across a range of multimodal datasets, namely MM-IMDb, UPMC Food101, and Hateful Memes.MM-IMDb serves for movie genre classification, incorporating both image and text modalities. Given the multigenre nature of movies, this dataset poses a multi-label classification challenge. The task involves predicting the set of genres a movie belongs to, utilizing either the image (movie poster), the text (movie plot), or a combination of both. UPMC Food-101 is constructed for the task of food classification, including both image and text modalities. It includes noisy image-text pairs retrieved from Google Image Search and aligns with the category structure of the ETHZ Food-101 dataset. HatefulMemes requires the identification of hate speech within memes using both image and text modalities. It is deliberately structured to foil unimodal models by introducing "benign confounders" thus necessitating effective multimodal analysis to achieve accurate classification.

| Dataset | Image | Text | Train | Val | Test |
|---|---|---|---|---|---|
| MM-IMDb | 25,959 | 25,959 | 15,552 | 2,608 | 7,799 |
| HateMemes | 10,000 | 10,000 | 8,500 | 500 | 1,500 |
| Food101 | 90,688 | 90,688 | 67,972 | – | 22,716 |

Table 1: Statistics of three multimodal downstream datasets.

**Input format.** For processing textual data, we utilize the `bert-base-uncased` tokenizer to convert input text into tokens. In scenarios where the text input is absent, we substitute it with an empty string as a dummy input (denoted by $\tilde{x}^{m_1}$). The maximum allowable input length for the text varies by dataset: 1024 tokens for MM-IMDb, 512 for UPMC Food-101, and 128 for Hateful Memes. For the image modality, following the preprocessing strategy, each input image is resized such that its shorter edge becomes 384 pixels, while the longer edge is constrained to not exceed 640 pixels, preserving the original aspect ratio. We divide each image into non-overlapping patches of size $32 \times 32$ pixels. If an image is not available, we replace it with a dummy image in which all pixel values are set to one, denoted as $\tilde{x}^{m_2}$.

## 2.More experiments

**Ablations of different rank of LoRA.** We also conduct experiments that adopt different LoRA rank. Although a larger rank can bring performance improvements, it also introduces more parameters. Here, we try different LoRA rank in order to find an optimal trade-off. As shown in the figure, when the rank is below 4, the model's performance on both the MM-IMDB and UPMCFood-101 datasets drops markedly. However, with a rank of 8, although the model still sees some performance gains, the degree of improvement is very limited—and the parameter count doubles compared to rank 4. Therefore, we ultimately choose rank = 4 as the default.

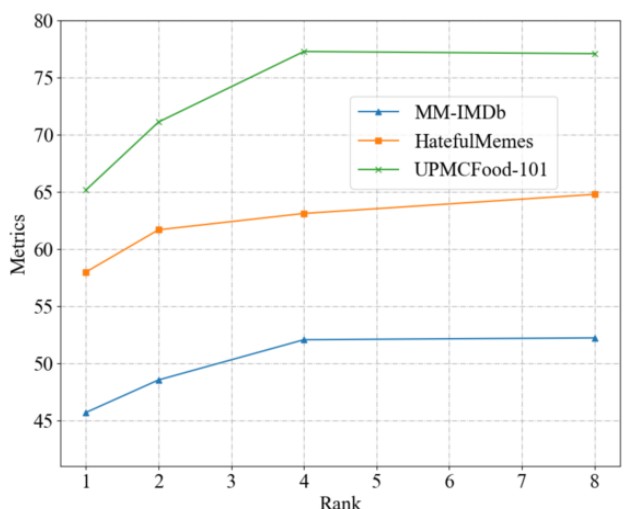

Figure 1: The impact of different LoRA.