# OpenReview forum: "MAIL: Mixture of LoRA Experts for Adaptive Incomplete Multimodal Learning"
_ICLR.cc/2026/Conference — ICLR 2026 Conference Withdrawn Submission_

### Official Review · Reviewer_Qvbj · 2025-10-20

**Soundness:** 2
**Presentation:** 1
**Contribution:** 1
**Rating:** 2
**Confidence:** 5

**Summary:**

This paper introduces MAIL (Mixture of LoRA Experts for Adaptive Incomplete Multimodal Learning), a method for handling missing modalities in multimodal tasks. The approach integrates a Mixture of Experts (MoE) architecture, composed of LoRA experts, into a frozen pre-trained ViLT backbone . The goal is to dynamically select experts based on the input data and its specific modality missingness pattern, thereby offering an instance-adaptive solution. While the paper addresses a relevant problem and shows strong results on its chosen benchmarks, it suffers from several major flaws that question its contribution and readiness for publication.

**Strengths:**

1. The paper tackles the important and practical challenge of missing modalities in real-world multimodal applications.
2. The authors conduct a comprehensive empirical evaluation on three datasets, demonstrating that their method outperforms a wide range of 11 baselines under the specified experimental conditions.

**Weaknesses:**

1. Use of an Outdated Backbone: The method is built and evaluated exclusively on ViLT, a model proposed over four years ago . The field of multimodal learning has advanced significantly since then with larger and more capable architectures. The reliance on an outdated backbone raises concerns about the relevance and generalizability of the findings to current state-of-the-art models.

2. Failure to Address Key Prior Work: the omission of highly relevant prior work. The paper fails to cite or compare against "Flex-MoE" (NeurIPS 2024), which also proposes using an MoE framework to handle missing modalities. This oversight directly challenges the originality and contribution of MAIL, as the core idea appears to have been previously explored.

3. Limited Novelty: The proposed method is a straightforward combination of two existing techniques: Mixture of Experts (MoE) and Low-Rank Adaptation (LoRA). While combining techniques can be valuable, the paper does not offer new insights into the fundamental problem of missing modalities.

4. Insufficient Experimental Depth: The evaluation, while broad, lacks depth. The central claim is that experts specialize and are selected adaptively, yet there is no qualitative analysis, such as visualizations of expert activation patterns, to support this. This makes it impossible to verify if the model is learning meaningful specializations or if the performance gains are due to other factors. The absence of simpler, strong baselines, such as training a separate LoRA adapter for each missingness combination, further weakens the experimental validation.

**Questions:**

See Weaknesses.

---

### Official Review · Reviewer_xKx1 · 2025-10-25

**Soundness:** 2
**Presentation:** 2
**Contribution:** 2
**Rating:** 2
**Confidence:** 4

**Summary:**

This paper proposes a LoRA-based missing-modality fine-tuning strategy to address incomplete learning in multimodal transformers. The main contributions are as follows:
(1) It introduces instance-specific prompts tailored for different samples.
(2) The number of prompts remains constant regardless of the number of modalities, making the method modality-irrelevant and scalable.
Also, extensive experiments validate the effectiveness.

**Strengths:**

+ **Integration of LoRA for Incomplete Multimodal Learning**: This work is the first to incorporate LoRA into incomplete multimodal learning within multimodal transformers, enabling efficient adaptation under missing-modality conditions.

+ **Modality-Independent Prompt Design**: At the design level, the number of prompts remains constant even as the number of modalities increases, ensuring both scalability and computational efficiency.

+ **Robust Performance**: Reliable performance across various multimodal missing settings.

**Weaknesses:**

+ **Lack of Novelty and Misplaced Contribution**: The authors claim that prior work in this domain adopts instance-agnostic prompts. However, both DCP [1] and RAGPT [2] already design dynamic prompts tailored to different missing-modality inputs. As a result, the novelty of this work is substantially weakened. Moreover, the authors’ description of the prior literature is inaccurate and misrepresents existing contributions.

+ **Design Concerns Regarding Experts and Modality Count**: Although the authors state that, at the design level, the number of experts is independent of the number of modalities, this claim does not fully hold in practice. When multiple modalities are involved, the number of experts can still increase.

+ **Insufficient Experiments on Efficiency**: Given that efficiency is a key motivation of this study, additional experiments are necessary to demonstrate that the proposed LoRA-based expert framework introduces less computational overhead compared to baseline models.

+ **Limited Modalities in Experimental Validation**: The experiments involve only a limited number of modalities. Since the authors argue that prompt complexity is independent of modality count, experiments with more modalities should be included. For example, in multimodal sentiment analysis tasks, missing-modality scenarios often involve three modalities, which would serve as a stronger validation setting.

[1] Deep Correlated Prompting for Visual Recognition with Missing Modalities, NeurIPS 2024.

[2] Retrieval-Augmented Dynamic Prompt Tuning for Incomplete Multimodal Learning, AAAI 2025.

**Questions:**

Please refer to the questions listed in the Weaknesses.

---

### Official Review · Reviewer_AbQf · 2025-10-27

**Soundness:** 2
**Presentation:** 2
**Contribution:** 2
**Rating:** 2
**Confidence:** 4

**Summary:**

This paper proposes Mixture of LoRA Experts for Adaptive Incomplete Multimodal Learning (MAIL). Specifically, the paper designs the LoRA-based Mixture of Experts and insert them into the pre-trained model to achieve adaptive incomplete multimodal learning. Experimental comparisons on three real-world datasets demonstrate that MAIL can effectively handle incomplete modality problems to 11 baselines.

**Strengths:**

1. This paper investigates different MoE disigns, as well as different routing strategies and expert allocation shapes, to investigate their impact under different modality-missing scenarios.
2. Experiments are conducted on three real-world datasets with 11 competitive baselines.

**Weaknesses:**

1. Typos, e.g., "Figuer" in line 087, "LoRa-based" in line 020, repeated "multimodel" misspellings in line 013 and Figure 2.
2. Only two modalities are studied, which limits the scope for incomplete multimodal learning.
3. Since the work focus on MoE+LoRA, it should compare against strong LoRA or MoE-structure baselines[1][2].
4. This paper states that MAIL enhances the model's inference performance and robustness, but I do not see explicit inference-time evaluations beyond accuracy-style merics.
5. The method is built on ViLT as the backbone, which is relatively old.

[1] Luo T, Lei J, Lei F, et al. Moelora: Contrastive learning guided mixture of experts on parameter-efficient fine-tuning for large language models[J]. arXiv preprint arXiv:2402.12851, 2024.
[2] Gao C, Chen K, Rao J, et al. MoLA: MoE LoRA with layer-wise expert allocation[C]//Findings of the Association for Computational Linguistics: NAACL 2025. 2025: 5097-5112.

**Questions:**

1. This paper appears to transfer the mixture of LoRA experts to the missing-modality setting. Please substantiate why mixture of LoRA experts outperform prompt-based tuning with deeper analysis, e.g. data-distribution perspective, layer-wise diagnostics on the ViLT backone, training stability and routing/expert specialization.
2. Can the approach extend to more modalities? The paper should evaluate robustness under richer missingness patterns.
3. Since the method centers on MoE+LoRA, the authors should compare against strong LoRA/adapter and MoE-structure fine-tuning baselines under identical settings and budgets.
4. The paper claims gains in inference performance and robustness but lacks efficiency evidence. Please report trainable parameter counts, memory usage, inference latency/throughput, and other relevant metrics (e.g., FLOPs), and consider an ablation over the number of experts and routing top-k to clarify accuracy–efficiency trade-offs.

---

### Official Review · Reviewer_cLAt · 2025-11-01

**Soundness:** 3
**Presentation:** 3
**Contribution:** 3
**Rating:** 4
**Confidence:** 4

**Summary:**

This paper proposes MAIL, a parameter-efficient fine-tuning method for handling missing modalities in multimodal tasks. Instead of static prompts, MAIL inserts a Mixture-of-Experts architecture composed of LoRA modules that are adaptively selected according to the missing-modality pattern and instance characteristics. Built upon ViLT and evaluated on several datasets, MAIL achieves superior or comparable results to competitive baselines while maintaining scalability and efficiency.

**Strengths:**

1. The primary strength is the conceptual shift from static, "instance-invariant" prompts to a dynamic, "instance-aware" MoE framework. The idea that the model can learn to route inputs based not just on the fact that text is missing, but on the content of the image itself, is a significant and valuable contribution.
2. The paper provides a compelling argument for its improved scalability. By making the parameter complexity dependent on a hyperparameter (number of experts) rather than the number of modalities $M$, it elegantly solves the $2^M - 1$ combinatorial explosion faced by methods like MAP. This is a strong engineering and architectural contribution.
3. The method is benchmarked against 11 baselines, including generation-based, joint learning, and prompt-based approaches, on three standard text-image datasets . MAIL variants achieve SOTA or highly competitive results in the majority of settings (Table 1), validating the effectiveness of the MoE-LoRA approach.

**Weaknesses:**

1. As an audio researcher, my primary concern is the lack of modality diversity. The paper's main motivation is solving the scalability issue as the number of modalities $M$ grows. However, all experiments are conducted on $M=2$ (text-image) datasets. The leap from $M=2$ (3 missingness cases) to $M=3$ (e.g., audio-video-text, 7 missingness cases) is non-trivial and represents the real testbed for the paper's scalability claim. I find it unconvincing to claim decoupled scalability without a single experiment where $M > 2$. The paper would be substantially stronger if it included results on a dataset like AudioSet, VGGSound, or a similar $M=3$ benchmark to prove that the MoE router can handle more complex missingness patterns (e.g., "only audio available," "video+text available").
2. The key claim is that MAIL is "instance-aware" while prior work is "instance-invariant". The mechanism for this is the MoE router $R(x)$. However, the experiments do not provide direct evidence that the router is actually using instance-specific features, rather than just learning a more complex set of "static-pattern-experts" (e.g., an expert for "text-missing-for-image," another for "image-missing-for-text"). I am not fully convinced the model is differentiating based on content. The paper needs a qualitative analysis to support this claim. For example, do two different images (e.g., "a cat" vs. "a car"), both with text missing, get routed to different expert combinations? A t-SNE visualization of input tokens, colored by their assigned expert, would be highly informative.
3. The ablation in Table 3 finds that MAIL-A (more experts at lower layers) performs best. This is an interesting finding, as it contradicts some other MoE-PEFT literature that suggests higher layers benefit more. The authors' explanation that lower layers "compensate for the missing information" is speculative. This finding needs more discussion. Is this a robust phenomenon, or a small statistical variance? I suspect this might be an artifact of using a frozen ViLT backbone, which forces the model to perform all adaptation at the earliest layers. This interaction is not explored.
4. While 11 baselines are included, they are all evaluated on the same text-image tasks. Given the paper's general claims about multimodal learning, the discussion and comparison feel insular to the vision-language domain. Some prior work on modality missingness exists in other areas (e.g., audio-visual fusion) [1, 2] which is not acknowledged.
5. The idea of MoE-based PEFT (e.g., AdaMix, MoKA) is not new. The novelty here lies in applying it to incomplete multimodal learning, but the paper could engage more critically with: (1) whether MAIL introduces any theoretical or algorithmic innovation beyond this application context; (2) how it differs from RAGPT beyond replacing dynamic prompts with LoRA experts, since they both use instance-aware adaptation mechanisms.

References

[1] https://arxiv.org/abs/1905.12681

[2] https://arxiv.org/abs/2305.01233

**Questions:**

See Weaknesses

---

### Note · Authors · 2025-11-12

I have read and agree with the venue's withdrawal policy on behalf of myself and my co-authors.